# Susceptibility to COVID-19 after High Exposure to Perfluoroalkyl Substances from Contaminated Drinking Water: An Ecological Study from Ronneby, Sweden

**DOI:** 10.3390/ijerph182010702

**Published:** 2021-10-12

**Authors:** Christel Nielsen, Anna Jöud

**Affiliations:** 1Division of Occupational and Environmental Medicine, Department of Laboratory Medicine, Lund University, SE-221 00 Lund, Sweden; anna.joud@med.lu.se; 2Department of Research and Education, Skåne University Hospital, SE-221 85 Lund, Sweden

**Keywords:** COVID-19, PFAS, drinking water contamination, public health, immunotoxicity

## Abstract

There is concern that immunotoxic environmental contaminants, particularly perfluoroalkyl substances (PFAS), may play a role in the clinical course of COVID-19 and epidemiologic studies are needed to answer if high-exposed populations are especially vulnerable in light of the ongoing pandemic. The objective was, therefore, to determine if exposure to highly PFAS-contaminated drinking water was associated with an increased incidence of COVID-19 in Ronneby, Sweden, during the first year of the pandemic. We conducted an ecological study determining the sex- and age-standardized incidence ratio (SIR) in the adult population relative to a neighboring reference town with similar demographic characteristics but with only background levels of exposure. In Sweden, COVID-19 is subject to mandatory reporting, and we retrieved aggregated data on all verified cases until 3 March 2021 from the Public Health Agency of Sweden. The SIR in Ronneby was estimated at 1.19 (95% CI: 1.12; 1.27). The results suggest a potential link between high PFAS exposure and susceptibility to COVID-19 that warrants further research to clarify causality.

## 1. Introduction

Concerns have been raised that immunotoxic environmental contaminants may interact with the severe acute respiratory syndrome coronavirus 2 (SARS-CoV-2) and perfluoroalkyl substances (PFAS) have been highlighted as a group of chemicals of specific concern [1].

PFAS are synthetic chemicals that have been used for numerous industrial purposes over decades and their extensive use has resulted in global contamination. PFAS bioaccumulate in humans and have endocrine-active properties, implying that they may mimic, block, or otherwise interfere with normal hormonal functioning and thus affect humoral immunity. There is evidence that two of the most widely used PFAS, perfluorooctanoic acid (PFOA) and perfluorooctane sulfonate (PFOS), are associated with immunosuppression that manifests through inverse associations with antibody titers after vaccination [2,3,4]. Further, there is some evidence to suggest that exposure is associated also with increased susceptibility to infections. Recently, the Agency for Toxic Substances and Disease Registry published a Statement on the potential intersection between PFAS exposure and COVID-19, recognizing that it is an important public health issue and that there is a need for research [5].

There is, to date, three studies that have investigated PFAS exposure in relation to COVID-19. Grandjean et al. [6] addressed the association between PFAS concentrations and *severity* of *disease* in an adult Danish population with confirmed COVID-19 and with background levels of PFAS exposure. They found no associations to plasma concentrations of PFOS, PFOA, and perfluorohexane sulfonic acid (PFHxS), but an odds ratio of 1.6 of increased COVID severity in individuals with detectable concentrations of perfluorobutanoic acid (PFBA; a PFAS with short elimination half-life) in relation to those with concentrations below the limit of detection. An ecological study from the Italian Veneto Region with PFAS exposure, dominated by PFOA, from contaminated drinking water reported a 1.6 higher COVID-19 *mortality* rate ratio in the contaminated district in relation to the rest of the region [7]. From a public health perspective, it is highly relevant to clarify whether PFAS also increases the susceptibility to SARS-CoV-2 because such knowledge would aid in identifying individuals at higher *risk of infection*. A recent case-control study of 160 participants from Shanxi and Shandong, Chinese provinces with both high PFAS exposure and high incidence of COVID-19, showed increased odds ratios of mild infection with increasing urine concentrations of PFOS, PFOA, and the sum of 12 PFAS [8]. Taken together, these studies provide some evidence that exposure to PFAS may indeed play a role in the clinical course of COVID-19, but further studies are needed to verify the findings. In particular, the need for epidemiological studies in larger cohorts and in highly exposed populations has been emphasized [1,6,8].

In Ronneby, Sweden, one-third of the population was exposed to drinking water contaminated with very high levels of primarily PFHxS and PFOS for decades. The objective of the study was to estimate the standardized incidence ratio (SIR) of COVID-19 in the Ronneby population during the first year of the pandemic relative to a neighboring town with background levels of exposure. The study uses mandatory-reported data from SmiNet (Public Health Agency of Sweden) on all registered cases of COVID-19, verified through free-of-charge testing by polymerase chain reaction (PCR).

## 2. Materials and Methods

### 2.1. Setting

In 2013, extremely high levels of several PFAS (i.e., sum PFAS > 10,000 ng/L) were discovered in the outgoing water from one of two public waterworks in Ronneby, a municipality in southern Sweden with a population size of 28,000 at the time. The source of the contamination was identified as leakage of aqueous film-forming firefighting foam from a military airfield near the water supply. It is not clear when the contamination reached the water table, but the firefighting foam was in use at the training site from the mid-1980s. The affected waterworks had supplied one-third of the households over time, primarily the Kallinge district, whereas two-thirds had received water from the unaffected waterworks with PFAS levels well below the current action limits [9,10]. The serum concentrations in the Ronneby population as a whole are substantially elevated, because those who did not receive the contaminated water at their home address have also worked or visited the parts of the town with contaminated water supplies.

The serum concentrations of the population based on biomonitoring of 3507 individuals (13% of the population) in 2014–2015 have been described in detail by Xu et al. [11]. In brief, individuals who had received only uncontaminated water at their home address had geometric mean serum concentrations of PFHxS, PFOS, and PFOA of 30, 40, and 3.5 ng/mL, respectively. These levels were 29, 8, and 2 times higher than the corresponding concentrations in the reference population of Karlshamn, a neighboring town with only background levels of exposure and similar population demographics as Ronneby (Figure 1). In individuals who had lived in the contaminated district after 2005, the serum concentrations of PFHxS, PFOS, and PFOA were 210, 239, and 13 ng/mL, respectively, with 95th percentiles of PFHxS and PFOS reaching almost 800 ng/mL.

### 2.2. Variables and Data Sources

The first case of COVID-19 in Blekinge County was confirmed on 11 March 2020. During the following spring, testing by means of PCR was primarily conducted to confirm or rule out SARS-CoV-2 infection in patients in need of healthcare, but also in infection tracing. Later, Blekinge County Council offered self-testing (PCR) for ongoing infection free of charge to adults and children older than 13 years, and with any symptom suggestive of COVID-19. Although the testing policy varied over the course of the pandemic, it was identical across the county at any given point in time, hence it was identical in Ronneby and the reference town.

COVID-19 is subject to mandatory reporting under the Communicable Diseases Act (2004:168) and verified cases are reported into SmiNet (Public Health Agency of Sweden). We retrieved aggregated data, stratified by sex and age, on population size and all laboratory-confirmed COVID-19 cases until 3 March 2021, in the adult population (≥20 years) with a home address in central Ronneby or central Karlshamn (reference population) according to the registered postal code. No further data on the characteristics of the cases were available.

Vaccination of the most vulnerable subgroups of the population (i.e., individuals living in care homes, with home health care or home care, and everyone aged 80 years or older) started in January 2021. The second vaccination phase, including individuals aged 65 years and older, started in March.

### 2.3. Patient and Public Involvement

Patients were not involved in the study. We used an administrative database covering all verified cases of COVID-19 but this was not community engaged research.

### 2.4. Statistical Analysis

We made two comparisons: (1) all districts of Ronneby together versus the reference and (2) the contaminated district versus the reference. We calculated crude and sex- and age-stratified cumulative incidences per 1000 individuals and indirect SIR using SAS version 9.4 (SAS Institute, Cary, NC, USA).

## 3. Results

The crude incidence per 1000 individuals was 68 in Ronneby (all districts), 66 in the contaminated district, and 56 in the reference town (Table 1).

The point estimates of the stratum-specific relative risk in Ronneby were generally above 1, except in females aged 20 to 29 years (Figure 2). We observed the highest point estimate in the age group 60–69 years, irrespective of sex.

There were more cases than expected in Ronneby and we estimated the SIR at 1.19 (Table 2). The SIR in the contaminated district was consistent with that of the town as a whole, although numerically slightly smaller and associated with greater uncertainty.

## 4. Discussion

In the midst of an ongoing pandemic, our intention was to provide a rapid first answer as to whether high PFAS exposure may be associated with increased susceptibility to COVID-19. Routinely collected data on an aggregate level could be made promptly available from the national disease-surveillance program and allowed us to estimate the SIR of COVID-19 in a context with a strong spatial exposure contrast.

We observed a SIR of verified COVID-19 in Ronneby, all districts together, of 1.19. The whole population of Ronneby, not only those residing in the contaminated district, has highly elevated PFAS concentrations compared with the general population. In 2014–2015, the geometric mean serum concentrations of PFHxS, PFOS, and PFOA were 114, 135, and 6.8 ng/mL [11]. These concentrations were 135, 35, and 5 times higher than those measured in the reference population from Karlshamn. To put these exposure levels into context, Ji et al. [8] performed their study in a population with median serum concentrations of PFOS and PFOA of 19 and 3.9 ng/mL. Yet, the authors reported clearly increased odds of COVID-19 infection with increasing exposure levels. Taken together with the current literature, the results continue to be suggestive of increased susceptibility to COVID-19 with higher PFAS exposure.

We did not find evidence of an even higher SIR in the contaminated district. Rather, the estimated SIR was reasonably comparable to that of Ronneby as a whole. This may be taken as weakening the evidence as there was no dose–response relationship despite the extreme serum concentrations in the contaminated district. However, the whole population that lived in Ronneby before the contamination was discovered does indeed have very high serum concentrations of several PFAS compared with the general population because people have been mobile within the town. As such, the results may well reflect that these exposure levels increased the susceptibility to COVID-19, but with no further increase as serum concentrations reached extreme levels. It would be desirable to validate this by studying dose–response relationships at lower exposure levels using individual measurements of exposure.

The serum concentrations of PFHxS, PFOS, and PFOA are clearly highest in the older population of Ronneby [11] and there was some evidence of higher SIR for COVID-19 in the older age groups, although not consistently. This could be interpreted as evidence for a dose–response pattern, though other age-related changes may be relevant. It would be of interest to explore whether a potential link pertains simply to exposure to the contaminated water over a longer period of time or if PFAS exposure may accelerate or otherwise modify age-related changes in immune function [12] so that the elderly population is rendered more susceptible to COVID-19.

### Strengths and Limitations

The ecological study design is associated with some inherent pros and cons. Ecological studies are helpful in generating hypotheses with respect to potential causes of public health concerns and, as such, offer a first step when approaching new research questions. When individual-level data is unavailable, as in our case, but aggregated data on exposure and outcome can be retrieved from registers, the ecological study is a time-efficient instrument in the epidemiologists’ toolkit.

On the limitation side, ecological studies are prone to bias and confounding. For instance, ascertainment bias may occur if ascertainment of the outcome differs between geographical locations. We do not consider this an issue in the current setting because the testing policy was identical across the county at all times and the similarities between the towns with respect to socioeconomic characteristics suggest that they would be equally inclined to be tested. We addressed confounding by age and sex through standardization but there may be other differences between the towns’ populations, such as education level, types of employment, and household crowding, that could, in part, explain the observed SIR. Still, older age and male sex are two of the strongest risk factors for COVID-19 infection [13] as well as for increased severity [14], and we ascertained a priori that the towns were comparable with respect to important demographic characteristics (Figure 1).

We used residential postal code at the time of diagnosis to classify participants in exposure groups, although the contaminated water district was provided with uncontaminated drinking water as soon as the contamination was discovered in late 2013. However, despite more than one elimination half-life having passed [15], the extreme exposure levels imply that the population’s serum concentrations will remain elevated for many years to come. Further, using the current address assumed that individuals had been sedentary within their residential district between the time of exposure and the current study period so there will be some misclassification of individuals, although the population average contrasts in PFAS exposure will be valid. The demographic characteristics of the towns did not provide reason to suspect that potential misclassification would have a systematic component and, if present, we assume it to be non-differential.

Another limitation could be that the analysis did not account for spatial clustering. However, we argue that potential clustering of cases may result from increased susceptibility to COVID-19 after the PFAS contamination, which would be in line with the hypothesis. Still, we were concerned that spatial clustering might drive the results if we had limited the study to the contaminated district only because of its relatively small geographic area and population size. We therefore estimated the SIR also for the town as a whole, which was justified based on the study by Xu et al. [11] showing that serum concentrations are elevated in all districts. We consider the similarity of the estimates to be reassuring that spatial clustering was not a severe issue, at least not between different districts within Ronneby. In this context, it should be noted that the spread of the mutated and more transmissible variants of the virus did not take off in Blekinge county until after the study period [16].

## 5. Conclusions

We observed a SIR of laboratory-confirmed COVID-19 of 1.19 in Ronneby where the population has been highly exposed to primarily PFHxS and PFOS from contaminated drinking water. Previous studies have found that PFAS is a potential immunotoxin and exposure is associated with reduced antibody response after vaccination. The results also suggest a potential link to COVID-19 susceptibility. The magnitude of the association is smaller than in some other studies but warrants further research to establish causality. In particular, given that the effect seems to occur at the lower exposure range, research is needed at lower and background levels which are more directly relevant to widespread public health concern. Future work should preferably be based on individual data to allow for more explicit exposure assessment and confounding control.

## Figures and Tables

**Figure 1 ijerph-18-10702-f001:**
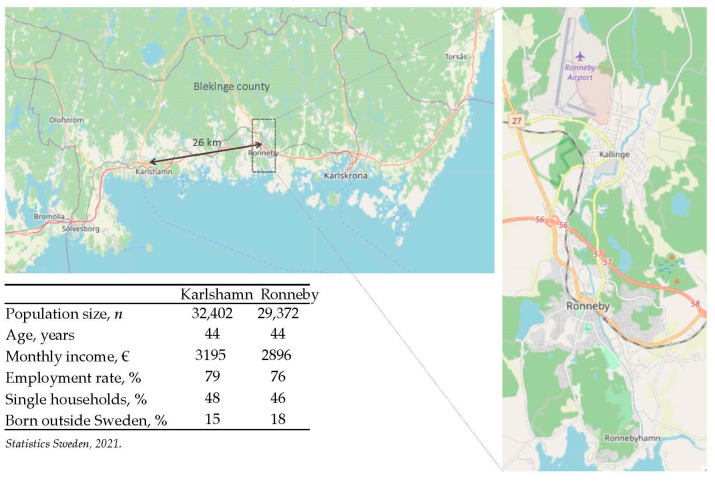
The location of Ronneby and Karlshamn in Blekinge county (**left**) and a magnification of the central parts of Ronneby showing the location of the high-exposed Kallinge district (**right**). Background map data: © OpenStreetMap contributors, CC BY-SA.

**Figure 2 ijerph-18-10702-f002:**
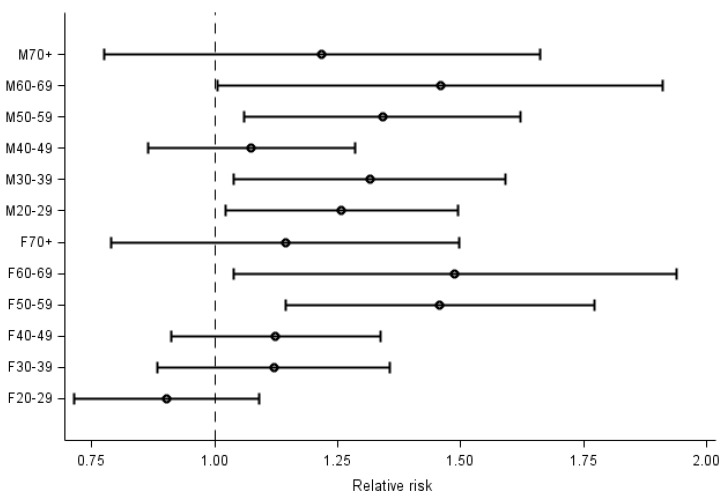
Stratum-specific relative risks and associated 95% confidence intervals of verified COVID-19 in Ronneby (all districts together) in relation to the reference town.

**Table 1 ijerph-18-10702-t001:** Verified cases of COVID-19 in Ronneby (all districts together), in the contaminated district, and in the reference town together with the cumulative incidence per 1000 individuals.

		Ronneby, All Districts			Contaminated District			Reference Town		
Sex	Age	Population (*n)*	Cases (*n*)	Cumulative Incidence	Population (*n)*	Cases (*n*)	Cumulative Incidence	Population (*n)*	Cases (*n*)	Cumulative Incidence
All	All	13,141	898	68	3635	239	66	11,059	622	56
Female	20–29	853	88	103	221	22	100	813	93	114
	30–39	922	86	93	277	23	83	756	63	83
	40–49	1000	107	107	286	30	105	693	66	95
	50–59	1045	83	79	305	25	82	899	49	55
	60–69	856	42	49	247	11	45	849	28	33
	70+	1800	40	22	447	9	20	1595	31	19
Male	20–29	1041	109	105	240	31	129	1009	84	83
	30–39	1127	87	77	300	19	63	869	51	59
	40–49	1079	100	93	285	27	95	707	61	86
	50–59	1149	87	76	357	20	56	903	51	56
	60–69	847	40	47	245	12	49	772	25	32
	70+	1422	29	20	425	10	24	1194	20	17

**Table 2 ijerph-18-10702-t002:** Standardized incidence ratios (SIR) and associated 95% confidence intervals (CI) of verified COVID-19 in Ronneby, all districts together, and in the contaminated district in relation to the reference town.

	Expected (*n*)	Observed (*n*)	SIR (95% CI)
Ronneby, all districts	752	898	1.19 (1.12; 1.27)
Contaminated district	206	239	1.16 (1.01; 1.30)

## Data Availability

The data underlying the results of the study is available from the Public Health Agency of Sweden (https://www.folkhalsomyndigheten.se/ (accessed on 11 October 2021)).

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
