# Peer review of "Susceptibility to COVID-19 after High Exposure to Perfluoroalkyl Substances from Contaminated Drinking Water: An Ecological Study from Ronneby, Sweden"

_ijerph, 2021, doi:10.3390/ijerph182010702_

Round 1

Reviewer 1 Report

I very much enjoyed your paper.  Here are some comments and suggestions for minor revisions and correction of typos.

Nielsen and Joud

29 and the extensively use

Correct the wording….I think they meant, “their extensive use.”

33 the two of the most widely

Correct the wording….”two of the most widely” (remove the from the sentence)

58 COVID-19, but further studies are needed

Insert a comma after COVID-19

107 Ronneby or central Karlshamn (reference) a

You have forgotten your citation here, please incorporate it into the paper and do not forget to include it in the bibliographic references.

109 and 110 2.3. Patient and public involvement 109

Patients were not involved in the study.   What about the public?  Public data bases were used but this was not community engaged research, correct?  Perhaps note that?

In your conclusions, please address the potential that your findings may be impacted because you are considering only one potentially immunotoxic substance among many possible additional exposures.  This actually makes you findings even more relevant and important to communicate.

Author Response

I very much enjoyed your paper.  Here are some comments and suggestions for minor revisions and correction of typos.

29 and the extensively use
Correct the wording….I think they meant, “their extensive use.”
AU: Indeed, this has been corrected.

33 the two of the most widely
Correct the wording….”two of the most widely” (remove the from the sentence).
AU: Corrected.

58 COVID-19, but further studies are needed
Insert a comma after COVID-19
AU: Comma inserted.

107 Ronneby or central Karlshamn (reference) a
You have forgotten your citation here, please incorporate it into the paper and do not forget to include it in the bibliographic references.
AU: This was actually supposed to mean that Karlshamn was our reference population. We have clarified that.

109 and 110 2.3. Patient and public involvement 109
Patients were not involved in the study.   
What about the public?  Public data bases were used but this was not community engaged research, correct?  Perhaps note that?
AU: SmiNet is not a public database in the sense that it is accessible to anyone. Rather, it is a kept by The Public Health Agency for administrative purposes, and researchers may access parts of it on request. We have added a sentence to clarify this.

In your conclusions, please address the potential that your findings may be impacted because you are considering only one potentially immunotoxic substance among many possible additional exposures.  This actually makes you findings even more relevant and important to communicate.
AU: Thank you for this very relevant comment. We have measured concentrations of other environmental exposures (other persistent organic pollutants and metals), both in the drinking water in both municipalities and in serum, and there were no indications that levels were elevated. Further, the PFAS concentrations in the Ronneby population are so extremely high that it would be very difficult to conclude anything about the impact of other exposures in this setting.

Reviewer 2 Report

Introduction - lines 30-32 - what is connection between endocrine active properties and immune function? please explain 

In introduction - line 33 - consider replacing term "consensus" with evidence

Introduction - although most readers generally familiar with COVID pandemic and risk factors it would be helpful to provide a few sentences on COVID pandemic in Sweden, what percent of adult population was infected? what were primary risk factors in Swedish context? Were PCR tests free of charge? Were PCR tests conducted in all individuals who were positive in self tests?  Were older adults vaccinated during the study period?  

Variables and data sources - 

  1. why was analysis limited to 20 and above? Are children in this region more highly exposed to PFAS (perhaps from breastmilk, contaminated foods?)
  2. Does the SmiNet database not provide any data on severity of disease?

In discussion of limitations, perhaps mention small population size. 

Author Response

Introduction - lines 30-32 - what is connection between endocrine active properties and immune function? please explain
AU: Hormones are involved in the regulation of humoral immunity and these processes may be disturbed by endocrine disruptors. A PFAS-related example is provided in the following section, i.e. reduced antibody titers after vaccination. We have now restructured the text so that these sentences are in the same section. The text reads better now, thank you!

In introduction - line 33 - consider replacing term "consensus" with evidence
AU: Changed accordingly.

Introduction - although most readers generally familiar with COVID pandemic and risk factors it would be helpful to provide a few sentences on COVID pandemic in Sweden, what percent of adult population was infected? what were primary risk factors in Swedish context? Were PCR tests free of charge? Were PCR tests conducted in all individuals who were positive in self tests?  Were older adults vaccinated during the study period?  
AU: Thank you for this excellent comment! In particular, the suggestion to add information on the vaccination pla was very valuable.

We have added information on PCR-tests being free of charge and COVID being subject to mandatory reporting in the Introduction. Further, we added additional information regarding vaccinations in the Materials & Methods. All self-tests were analyzed by PCR, and this has been clarified. There is no information on risk factors that is specific to Sweden, the pattern is the same here as in other countries (primarily age and sex as stated in the Discussion). We found it a bit difficult to squeeze in the cumulative incidence of COVID in the Swedish population, as the paper jumps straight into the Ronneby setting and the pandemic developed very differently in different parts of the country. We hope that the adjustments/additions that we have made provide some context of COVID in Sweden.

Variables and data sources - 

  1. why was analysis limited to 20 and above? Are children in this region more highly exposed to PFAS (perhaps from breastmilk, contaminated foods?)
    AU: This decision was based on the fact that children were not tested at all in Sweden during a large proportion of the pandemic’s first year. There were consequently few cases (lack of statistical power) and, more importantly, those that were reported were assumed not to be representative of the underlying population.

  2. Does the SmiNet database not provide any data on severity of disease?
    AU: No, unfortunately not. SmiNet only contains date of symptom onset, test results, and demographic characteristics. Information regarding severity can be retrieved from medical journals, and we are currently undertaking such a study.

In discussion of limitations, perhaps mention small population size.
AU: We agree that we have studied a relatively small dataset. However, we retrieved data on all cases that occurred during the pandemic’s first year in the total populations of Ronneby and Karlshamn. The value of the study design lies in the very high exposure levels in Ronneby, and the substantial exposure contrast relative to Karlshamn, rather than in the sample size.

With respect to the Limitations section, we discuss sample size in relation to spatial clustering in the last paragraph, together with the measure that we took to reduce its impact on the results (i.e. that we not only estimated the SIR in the exposed district, but also for the whole town).

One way to increase sample size would be to include cases over a longer time period. However, there is speculation that the association between PFAS exposure and COVID might be different at different phases of the pandemic, and that some kind of harvesting effect would be at play. We therefore think that the first year of the pandemic is particularly interesting to study, although at the cost of relatively few cases and hence less statistical power.

Following the results of this study, we are planning to revisit the research question based on individual-level data from registries and medical records. This is a slow process though.